# Where do Models go Wrong? Parameter-Space Saliency Maps for Explainability

## Abstract

Conventional saliency maps highlight input features to which neural network predictions are highly sensitive. We take a different approach to saliency, in which we identify and analyze the network parameters, rather than inputs, which are responsible for erroneous decisions. We first verify that identified salient parameters are indeed responsible for misclassification by showing that turning these parameters off improves predictions on the associated samples, more than pruning the same number of random or least salient parameters. We further validate the link between salient parameters and network misclassification errors by observing that fine-tuning a small number of the most salient parameters on a single sample results in error correction on other samples which were misclassified for similar reasons – nearest neighbors in the saliency space. After validating our parameter-space saliency maps, we demonstrate that samples which cause similar parameters to malfunction are semantically similar. Further, we introduce an input-space saliency counterpart which reveals how image features cause specific network components to malfunction.

## 1 Introduction

With the widespread deployment of deep neural networks in high-stakes applications such as medical imaging (Kang et al., 2017), credit score assessment (West, 2000), and facial recognition (Deng et al., 2019), practitioners need to understand why their models make the decisions they do. In fact, "right to explanation" legislation in the European Union and the United States dictates that relevant public and private organizations must be able to justify the decisions their algorithms make (United States Congress Senate Committee on Banking and Housing and Urban Affairs, 1976; European Commission, 2018). Diagnosing the causes of system failures is particularly crucial for understanding the flaws and limitations of models we intend to employ.

Conventional saliency methods focus on highlighting sensitive pixels (Simonyan et al., 2014) or image regions that maximize specific activations (Erhan et al., 2009). However, such maps may not be useful in diagnosing undesirable model behaviors as they do not necessarily identify areas that specifically cause bad performance since the most sensitive pixels may not be the ones responsible for triggering misclassification.

We develop an alternative approach to saliency which highlights network parameters that influence decisions rather than input features. These parameter saliency maps yield a number of useful analyses:

- Nearest neighbors in parameter saliency space share common semantic information. That is, samples which are misclassified for similar reasons and cause similar parameters to malfunction are semantically similar.

- By first identifying the network parameters responsible for an erroneous classification, we can then visualize the image regions that interact with those parameters and trigger the identified misbehavior.

- We verify that identified salient parameters are indeed responsible for misclassification by showing that turning these parameters off improves predictions on the associated samples, more than pruning the same number of random or least salient parameters.

- We further validate the link between salient parameters and network misclassification errors by observing that fine-tuning a small number of the most salient parameters on a single

sample results in error correction on other samples which were misclassified for similar reasons.

After carefully delineating our methodology and experimentally validating the meaningfulness of our parameter saliency maps, we showcase the practical utility of this paradigm as an explainability tool with a case study in which we are able to uncover a neural network's reliance on a spurious correlation which causes interpretable failures.

## 1.1 RELATED WORK

**Neural network interpretability and parameter importance.** A major line of work in neural network interpretability focuses on convolutional neural networks. Works visualizing, interpreting, and analysing feature maps (Zeiler & Fergus, 2014; Yosinski et al., 2015; Olah et al., 2017; Mahendran & Vedaldi, 2015) provide insight into the role of individual convolutional filters. These methods, together with other approaches for filter explainability (Bau et al., 2017; Zhou et al., 2018; 2019) find that individual convolutional filters often are responsible for specific tasks such as edge, shape, and texture detection.

The idea of measuring neural network parameter importance has been studied in multiple contexts. Notions of neuron and parameter importance have been used for AI explainability (Srinivas & Fleuret, 2019; Selvaraju et al., 2017; Morcos et al., 2018; Shrikumar et al., 2017; Shrikumar et al.), manipulating model behavior (Bau et al., 2018), and parameter pruning (Abbasi-Asl & Yu, 2017; Liu & Wu, 2019).

**Input space saliency maps.** A considerable amount of literature focuses on identifying input features that are important for neural network decisions. These methods include using deconvolution approaches (Zeiler & Fergus, 2014) and data gradient information (Simonyan et al., 2014). Several works build on these ideas and propose improvements such as Integrated Gradients (Sundararajan et al., 2017), SmoothGrad (Smilkov et al., 2017), and Guided Backpropagation (Springenberg et al., 2015) which result in sharper and more localized saliency maps. Other approaches focus on the use of class activation maps (Zhou et al., 2016) with improvements incorporating gradient information (Selvaraju et al., 2017) and more novel approaches to weighting the activation maps (Wang et al., 2020). In addition, various saliency methods are based on manipulating the input image (Fong & Vedaldi, 2017; Zeiler & Fergus, 2014). Another line of work is aimed at evaluating the effectiveness of saliency maps (Adebayo et al., 2018; Alqaraawi et al., 2020).

Although extensive work studies how different regions of *images* affect a network's predictions, limited work (Srinivas & Fleuret, 2019) aims to distinguish important network *parameters*. Our work combines the ideas of saliency maps and parameter importance and evaluates saliency directly on model parameters by aggregating their absolute gradients on a filter level. We leverage the resulting parameter saliency profiles as an explainability tool and develop an input-space saliency counterpart which highlights image features that cause specific filters to malfunction to study the interaction between the image features and the erroneous filters.

## 2 METHOD

It is known that different network filters are responsible for identifying different image properties and objects (Zeiler & Fergus, 2014; Yosinski et al., 2015; Olah et al., 2017; Mahendran & Vedaldi, 2015). This motivates the idea that mistakes made on wrongly classified images can be understood by investigating the network parameters, rather than only the pixels, that played a role in making a decision. We develop parameter-space saliency methods geared towards identifying and analyzing neural network parameters that are responsible for making erroneous decisions. Central to our method is the use of gradient information of the loss function as a measure of parameter sensitivity and optimality of the network at a given point in image space.

## 2.1 PARAMETER SALIENCY PROFILE

Let $x$ be a sample in the validation set $D$ with label $y$, and suppose a trained classifier has parameters $\theta$ that minimize a loss function $\mathcal{L}$. We define the *parameter-wise* saliency profile of $x$ as a vector

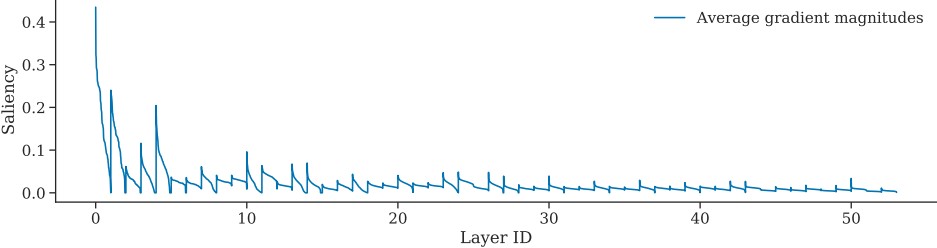

Figure 1: **Filter-wise parameter saliency profile.** ResNet-50 filter-wise saliency profile (without standardization) averaged over samples in ImageNet validation set. The filter saliency values in each layer are sorted in descending order, and each layer's saliency values are concatenated. The layers are displayed left-to-right from shallow to deep and have equal width on x-axis.

$s(x, y)$ with entries $s(x, y)_i := |\nabla_{\theta_i}\mathcal{L}_\theta(x, y)|$, the magnitudes of the gradient of the loss with respect to each model parameter. Because the gradients on *training* data for a model trained to convergence are near zero, it is important to specify that $D$ be a validation, or holdout, set. Intuitively, a larger gradient norm at the point $(x, y)$ indicates a greater inefficiency in the network's classification of sample $x$, and thus each entry of $s(x, y)$ measures the suboptimality of individual parameters.

**Aggregation of parameter saliency.** Convolutional filters are known to specialize in tasks such as edge, shape, and texture detection (Yosinski et al., 2015; Bau et al., 2017; Olah et al., 2017). We therefore choose to aggregate saliency on the filter-wise basis by averaging the gradient magnitudes of parameters corresponding to each convolutional filter. This allows us to isolate filters to which the loss is most sensitive (*i.e.* those which, when corrected, lead to the greatest reduction in loss).

Formally, for each convolutional filter $\mathcal{F}_k$ in the network, consider its respective index set $\alpha_k$, which gives the indices of parameters corresponding to the filter $\mathcal{F}_k$. The *filter-wise* saliency profile of $x$ is defined to be a vector $\overline{s}(x, y)$ with entries

$$\overline{s}(x, y)_k := \frac{1}{|\alpha_k|} \sum_{i \in \alpha_k} s(x, y)_i, \tag{1}$$

the parameter-wise saliency profile aggregated by averaging on the filter level.

**Standardizing parameter saliency.** Figure 1 exhibits the ResNet-50 (He et al., 2016) filter-wise saliency profile averaged over the ImageNet (Deng et al., 2009) validation set, where filters within each layer are sorted from highest to lowest saliency. One clear observation is the difference in the scale of gradient magnitudes – shallower filters are more salient than deeper filters. This phenomenon might occur for a number of reasons. First, early filters encode low-level features, such as edges and textures, which are active across a wide spectrum of images. Second, typical networks have fewer filters in shallow layers than in deep layers, making each individual filter more influential at shallower layers. Third, the effects of early filters cascade and accumulate as they pass through a network.

To isolate filters that uniquely cause *erroneous* behavior on particular samples, we find filters that are abnormally salient for a sample, $x$, but not for others. That is, we further standardize the saliency profile of $x$ with respect to all filter-wise saliency profiles of $D$.

Formally, let $\mu$ be the average filter-wise saliency profile across all $x \in D$, and let $\sigma$ be an equal-length vector with the corresponding standard deviation for each entry. We use these statistics to produce the standardized filter-wise saliency profile as follows:

$$\hat{s}(x, y) := \frac{|\overline{s}(x, y) - \mu|}{\sigma}. \tag{2}$$

The resulting tensor $\hat{s}(x, y)$ is of length equal to the number of convolutional filters in the network, and we henceforth call it the saliency profile for sample $x$. By standardizing saliency profiles, we create a saliency map that activates when the importance of a filter is unusually strong relative to other samples in the dataset. This prevents the saliency map from highlighting filters that are uniformly important for all images, and instead focuses saliency on filters that are uniquely important and serve

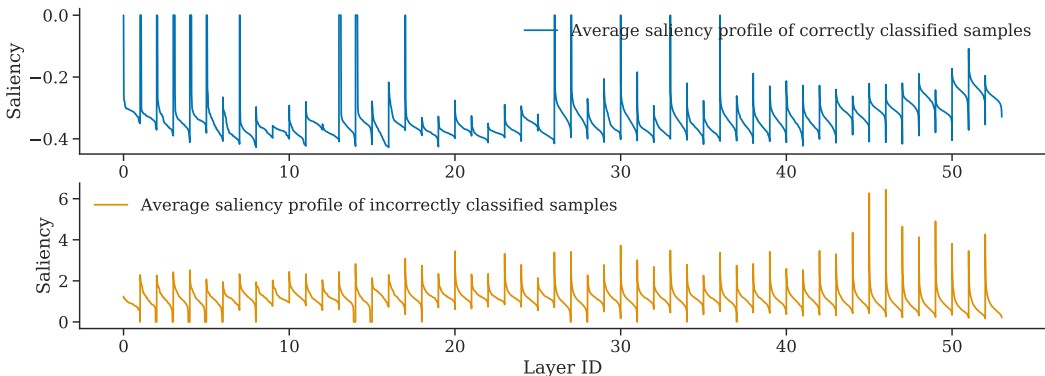

Figure 2: **Standardized filter-wise saliency profiles, correctly vs incorrectly classified samples.**
Top: Standardized saliency profiles averaged over correctly classified samples in the ImageNet
validation set. Bottom: Standardized saliency profiles averaged over incorrectly classified samples
in the ImageNet validation set. On both panels, the filter saliency values in each layer are sorted
in descending order, and each layer's saliency values are concatenated. The layers are displayed
left-to-right from shallow to deep and have equal width on x-axis. Both profiles are generated on
ResNet-50.

an image-dependent role. In the rest of the paper, unless explicitly noted otherwise, we use $\hat{s}(x, y)$
and refer to it as *parameter saliency*.

**Incorrectly classified samples are more salient.** Empirically, we observe the saliency profiles of
incorrectly classified samples exhibit, on average, greater values than those of correctly classified
examples. This bolsters the intuition that salient filters are precisely those malfunctioning — if
the classification is correct, there should be few malfunctioning filters or none at all. Moreover,
we see deeper parts of the network appear to be most salient for the incorrectly classified samples
while earlier layers are often the most salient for correctly classified samples. An example of
these behaviors for ResNet-50 is shown in Figure 2 which presents standardized filter-wise saliency
profiles averaged over the correctly and incorrectly classified examples from the ImageNet validation
set. Additionally, we note the improved relative scale of the standardized saliency profile across
different layers compared to the absolute gradient magnitudes in Figure 1. Saliency profiles for
other architectures could be found in Appendix A. Henceforth, we will focus specifically on saliency
profiles of *misclassified* samples in order to explore how neural networks make mistakes.

## 2.2 INPUT-SPACE SALIENCY FOR VISUALIZING HOW FILTERS MALFUNCTION

The parameter saliency profile allows us to identify filters that are most responsible for mistakes and
erroneous network behavior. In this section, we develop an input-space counterpart to our parameter
saliency method to understand which features of the image affect the saliency of particular filters.
Geiping et al. (2020) show that the gradient information of a network is invertible, providing a
link between input space and parameter saliency space. This work, along with existing input-space
saliency map tools (Simonyan et al., 2014; Springenberg et al., 2015; Smilkov et al., 2017; Zhou
et al., 2016; Selvaraju et al., 2017), inspires our method.

Given a parameter saliency profile $\hat{s} = \hat{s}(x, y)$ for an image $x$ with label $y$, our goal is to highlight
the input features that drive large filter saliency values. That is, we would like to identify image pixels
altering which can make filters more salient. To this end, we first select some set $F$ of the most salient
filters that we would like to explore. Then, we create a boosted saliency profile $s'_F$ by increasing the
entries of $\hat{s}$ corresponding to the chosen filters $F$ (e.g., multiplying by a large constant). Now, we can
find pixels that are important for making the chosen filters $F$ more salient and, equivalently, making
the filter saliency profile $\hat{s}(x, y)$ close to the boosted saliency profile $s'_F$ by taking the following
gradients:

$$M_F = |\nabla_x D_C(\hat{s}(x, y), s'_F)|, \tag{3}$$

where $D_C(\cdot, \cdot)$ is cosine distance.

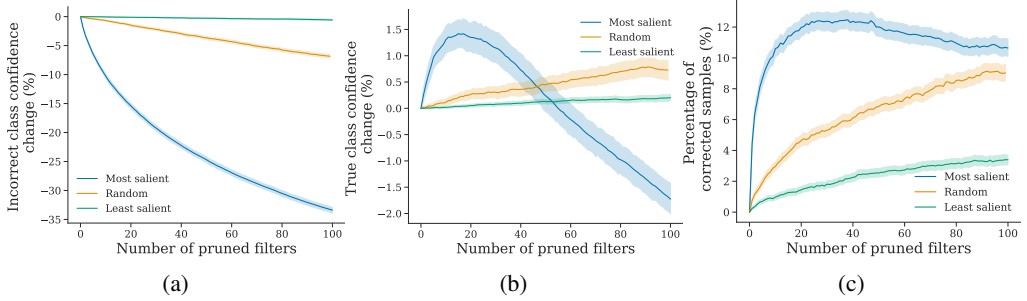

Figure 3: **Effect of turning salient filters off.** (a) Change in incorrect class confidence score. (b) Change in true class confidence score. (c) Percentage of samples that were corrected as the result of pruning filters. These trends are averaged across all images misclassified by ResNet-50 in the ImageNet validation set. The error bars represent 95% bootstrap confidence intervals.

The resulting input saliency map $M_F$ contains input features (pixels) that affect the saliency of the chosen filters $F$ the most.

## 3 EXPERIMENTS

In this section, we aim to validate the meaningfulness of our parameter saliency method. First, we show on the dataset level that turning salient parameters off improves predictions on the associated samples thus verifying that the salient parameters are indeed responsible for misclassification. We then find that samples which cause similar filters to malfunction are semantically similar. We also show on the dataset level that fine-tuning a small number of the most salient parameters on a single sample results in error correction on other samples which were misclassified for similar reasons. We then use our input-space saliency technique in conjunction with its parameter-space counterpart as an explainability tool to explore how neural networks make mistakes and how salient filters interact with visual input features.

We evaluate our saliency method in the context of image classification on CIFAR-10 (Krizhevsky, 2009) and ImageNet (Deng et al., 2009). Images we use for visualization, unless otherwise specified, are sampled from ImageNet validation set. Throughout the experiments, we use a pre-trained ResNet-18 classifier (He et al., 2016) on CIFAR-10 and a pre-trained ResNet-50 on ImageNet. Both models are trained in a standard fashion on the corresponding dataset[1][2].

### 3.1 PRUNING SALIENT FILTERS

We begin validating the meaningfulness of our parameter-space saliency maps by turning off the malfunctioning filters which cause misclassification of the associated image. We prune away the most salient convolutional filters (i.e., filters identified by our method as malfunctioning). In order to remove the influence of a particular salient filter, we zero out the filter weights and also the biases of the associated batch normalization layers. This procedure guarantees that the corresponding input feature map to the next convolutional layer is always zero.

Remarkably, we find that this simple procedure improves the network's behavior on the associated samples. In particular, we gradually increase the number of pruned most salient filters and track three metrics: the change in the incorrect class confidence, the change in the true class confidence, and the percentage of the samples that flip their label to the correct class. In every case, we compare pruning the most salient filters against pruning the same number of random filters and the least salient filters. These experiments are performed on the dataset level: we average the trends across all misclassified images in the ImageNet validation set.

As shown in Figure 3, pruning the most salient filters is significantly more effective for decreasing the incorrect class confidence than random or least salient filters. Specifically, gradually pruning the

---

[1]https://github.com/kuangliu/pytorch-cifar (under MIT license)
[2]https://github.com/pytorch/vision (under BSD 3-Clause License)

top 100 salient filters achieves up to 30% drop in the incorrect class confidence score while pruning random filters yields only about 7% decrease. We also note that pruning the least salient filters does not produce any effect on the incorrect class confidence.

We repeat the same experiment with the true class confidence and observe that the highest true confidence gain occurs when we prune around 20 most salient filters. Pruning enough salient filters eventually leads to a gradual decrease in the true class confidence. We note that this behavior is expected since we are destroying, not correcting, the inference power of all of the most sensitive filters for an image, some of which may be essential for inference. Finally, pruning random filters provides a much slower increase in the true confidence class while the least salient filters again do not produce a significant effect.

In addition, we count the number of images that were corrected as a result of pruning and find that pruning around 30 most salient filters results in the best correct classification rate of 12%. Similar to the true class confidence, the trend decreases beyond this point. Pruning the random filters increases the percentage of corrected samples at a much slower rate and does not perform better than the most salient filters when pruning up to 100 filters. Notably, pruning the least salient filters manages to correct a nontrivial number of samples but still much smaller than pruning random filters.

Given the trends in panels (b) and (c) of Figure 3, and given that pruning is a coarse tool for fixing misbehavior, we explore the natural idea of correcting the most salient filters instead of removing them altogether in our fine-tuning experiments in Section 3.3.

## 3.2 NEAREST NEIGHBORS IN PARAMETER SALIENCY SPACE

We validate the semantic meaning of our saliency profiles by clustering images based on the cosine similarity of their profiles. In this section, we present visual depictions of a nearest neighbor search among all images in the ImageNet validation set. We also conduct this analysis on CIFAR-10 images, and this can be found in Appendix A.

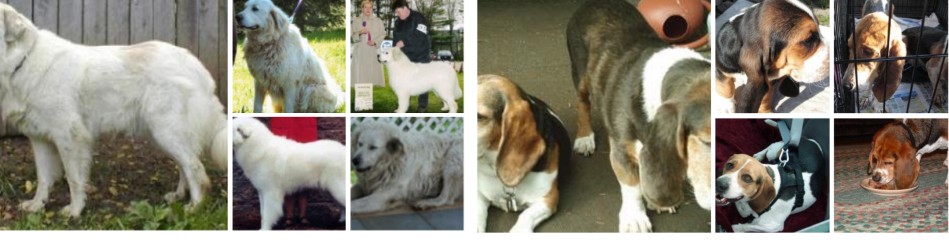

(a) Great pyrenees ↔ Kuvasz    (b) Basset hound ↔ Beagle

Figure 4: **Examples of nearest neighbors in parameter saliency space (from ImageNet)**.

We find that the nearest neighbors of misclassified images in saliency space are mostly other misclassified images from the same pair of predicted and true classes but possibly in reverse order. For example, in Figure 4, the reference image in (a) is a great Pyrenees misclassified as kuvasz, and the 4 images with the most similar profiles exhibit either the same misclassification or the reverse (i.e., kuvasz misclassified as great Pyrenees). Intuitively, the common salient parameters across these neighbors are those which are important for discriminating between the two classes in question but are not well-tuned for this purpose.

Note that we find the concept of "being similar" in parameter saliency space to be different from the one in image space. The nearest neighbors we find are often not similar in a pixel-wise sense, but rather they are similar in their reason for causing misclassification. For example, images in Figure 4 (b) are beagles mistaken by a network for basset hounds and vice versa. We find that these pictures are either taken from a high angle or do not include the dog's legs, making the leg length, a major distinction between the two breeds, indistinguishable from the picture. We include more example images along with their nearest neighbors in Appendix A.

In addition, we compute nearest neighbors when only considering filters in a specific range of layers in order to visualize the types of misbehavior triggered by network components (filters) at various network depths. We search for similar images using parameter saliency in the shallow and deep

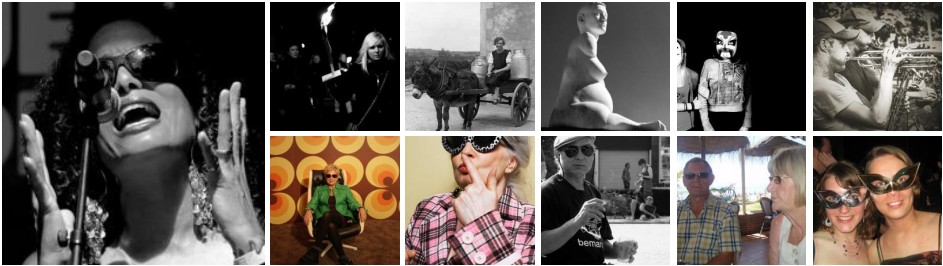

Figure 5: **Neighbors in parameter saliency space found using only early or only deep layers.** The reference image is in the first column. Images in the top row resemble the reference image in the saliency on early layers of VGG-19, and images in the bottom row are found using deeper layers.

layers of a VGG-19 network (Simonyan & Zisserman, 2015), which we divide into the shallow and deep parts that respectively occur up to and after layer `relu4_1`. The top row of figure Figure 5 shows neighbors found using shallow parameters, which share basic image attributes such as color histogram, while images in the bottom row share more abstract similarities.

### 3.3 CORRECTING MISTAKES BY FINE-TUNING SALIENT FILTERS

To validate whether salient filters are more responsible for the erroneous behavior of neural networks, we show that updating salient filters alone is sufficient for correcting the mistakes made by a neural network. In this experiment, for a pre-trained image classification network, we fine-tune it for one step on a single image for which the network makes the wrong prediction. We restrict the number of tunable filters to be no more than 1.0% of the total number of filters in a network, and we update the chosen filters by taking one step of gradient descent with a fixed step size. To validate the effectiveness of optimizing salient filters, we compare it with two other choices of tunable filters: the least salient filters and random filters. For a more general evaluation, we use images from the ImageNet validation set that are misclassified by a ResNet-50, making up to over 10,000 samples. We evaluate the effect of fine-tuning on each of these images independently.

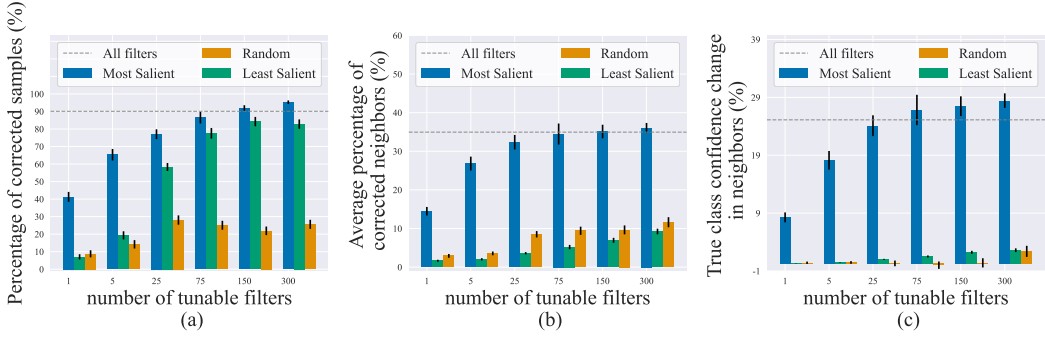

Figure 6: **Effect of updating a small number of filters.** (a) Percentage of samples that are corrected after fine-tuning. (b) Average percentage of nearest neighbors that are also corrected after fine-tuning. (c) Average change in the confidence score of the true class among nearest neighbors. The horizontal line in each plot is the effect of updating the entire network.

In Figure 6, we compare the average performance of our three choices of tunable filters under three evaluation metrics. For a given sample image and a set of tunable filters, an update step is considered to be effective if the updated network corrects its mistake on the sample image. In addition, it is more useful if the updated network also corrects its mistake on other images that resemble the sample image but are not seen during fine-tuning.

First, by inspecting the percentage of samples that are corrected after fine-tuning (Figure 6 (a)), we find that updating 150 salient filters (∼0.6% of total filters) can achieve the same result as updating the entire network. The second and third metrics evaluate the effect on the nearest neighbors of the

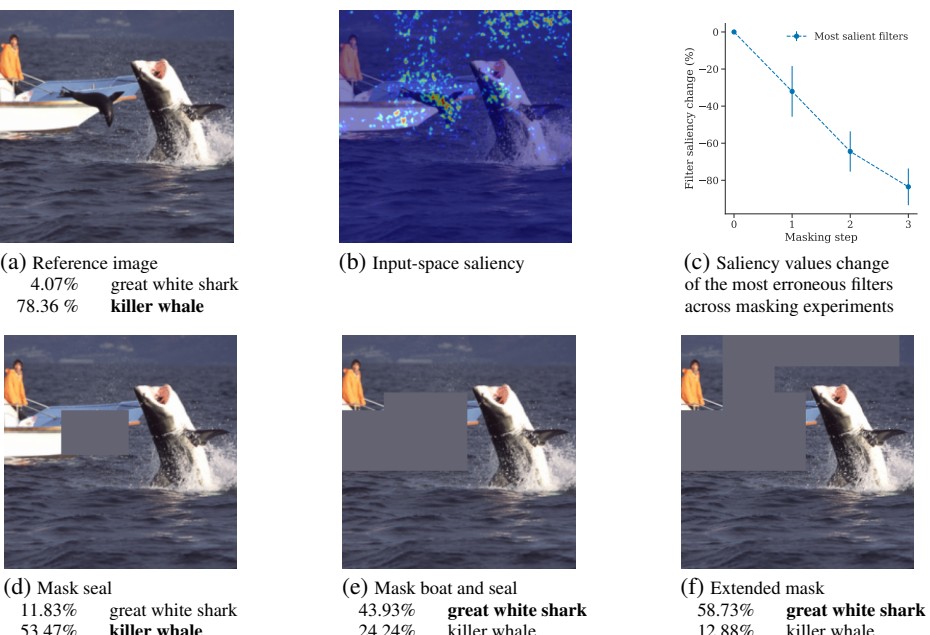

Figure 7: **Interaction between input features and salient filters.** (a) Reference image of "great white shark" misclassified by ResNet-50 as "killer whale" with confidence scores. (b) Input-space saliency visualization. Pixels that cause the top 10 salient filters to have high saliency. (c) Change in saliency values of the erroneous filters across masking experiments. The vertical bars represent the standard deviation of the change across 10 most salient filters. (d)-(f) Masking experiments.

training sample (Figure 6 (b),(c)). We find nearest neighbors for each training sample through the process introduced in Section 3.2, where we limit the search scope exclusively to other misclassified images. Note that for a given training sample, its nearest neighbors are not involved in our one-step single sample fine-tuning process. By tracking model predictions and true class confidence scores among the 10 nearest neighbors of each sample, we find that fine-tuning salient filters is significantly more effective than other choices. Results in Figure 6, (b) and (c), also imply that the nearest neighbors found using our method are the images that are wrong for similar reasons and that they can be corrected altogether by only updating the salient filters on a single image. We note that we do not propose a new pruning or fine-tuning method. Rather, we use these experiments to verify that the salient filters are indeed responsible for misclassification.

### 3.4 INPUT FEATURES THAT CAUSE FILTERS TO MALFUNCTION: A CASE STUDY

We consider a case study of an image misclassified by ResNet-50 as "killer whale" (Figure 7(a)). The correct label of the image is "great white shark". Our goal is to study the interaction between the most salient filters and input features. We first identify filters most responsible for misclassification by computing the filter saliency profile and visualize parts of the image that drive the high saliency values for those filters using the input-space saliency counterpart (Section 2.2).

Panel (b) of Figure 7 presents our image-space visualization, which depicts the causes of misbehavior for the ten most salient filters – the pixels that trigger misbehavior in these filters are highlighted. For example, we see that the seal and boat are both triggers. One natural hypothesis is that the seal looks like a killer whale to the network and is the source of the classification error. We test this hypothesis by masking out the seal (Figure 7 (d)) . However, although the probability of "killer whale" goes down and the probability of the correct class increases, the network still misclassifies the image as "killer whale".

Now, if we mask out exactly the most salient areas of the image according to our visualization (see Figure 7 (b), (e)), the network manages to flip the label of the image and classify it correctly. If we extend our mask to the less pronounced, but still salient, areas of the image as in Figure 7

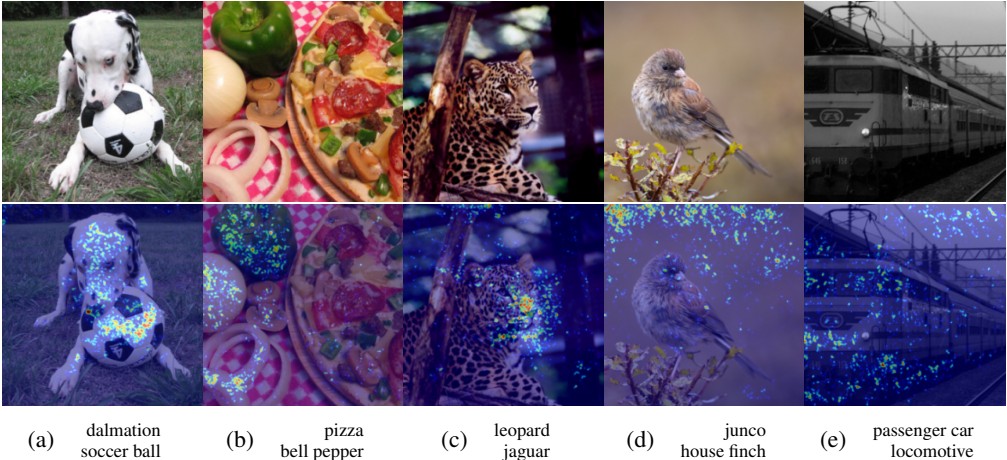

| (a) | dalmation
soccer ball | (b) | pizza
bell pepper | (c) | leopard
jaguar | (d) | junco
house finch | (e) | passenger car
locomotive |

Figure 8: **Different types of network mistakes.** All of the presented images are misclassified by ResNet-50. The correct class label is specified in the top row and the incorrect class label – in the bottom row of the subcaption on each panel. (a)-(b) The target object is confused with another object in the image. (c) A regular mistake. The salient pixels are focused on the target object features which confuse the network. (d) Background features confuse the network. (e) An example of a noisy label where the network is "more correct" than the target label. These are examples where masking top 5% of the salient pixels corrects the misclassification.

(f), we observe that the correct class confidence increases even more while the probability of the incorrect "killer whale" label further decreases. Additionally, we find that masking out the non-salient parts of the image results in even worse misclassification confidence than that of the original image (see Appendix A). In order to further investigate the effect of the salient region, we pasted it from this image onto other great white shark images (see Appendix A) and observed that this drives the probability of "killer whale" up for 39 out of 40 examples of great white sharks from the ImageNet validation set with an average increase of 3.75%.

Our experiments suggest that secondary objects in the image are associated with the misclassification. However, we see that the erroneous behavior of the model does not just stem from classifying a non-target object in the image. It is possible that the model correlates the combination of sea creatures (e.g. a seal) and man-made structures (e.g. a boat) with the "killer whale" label. We note that images of killer whales in ImageNet often have man-made structures which look similar to the boat (see Appendix A for examples of other "killer whale" images).

Finally, at each step of our masking experiments, we recompute the saliency values of the originally chosen 10 filters (i.e. the filters that caused erroneous behavior on the reference image). From Figure 7 (c), we observe that as we mask out the input features according to our input-saliency, the saliency values of those filters decrease gradually and reach an 80% drop, confirming that highlighted regions indeed drive the high saliency of the chosen filters.

More visualizations of input space saliency showcasing different illustrative examples of neural network mistakes can be found in Figure 8. For a thorough discussion of mistake categories we identify using our saliency method, we refer to Appendix A.

## 4 DISCUSSION

Numerous applications demand that practitioners be able to understand the decisions their models make, especially when those decisions are incorrect. Existing methods for explainability focus on locating the input regions to which the network's output is sensitive or on associating network components with specific roles. In contrast, we develop a framework for finding the exact filters which are responsible for faulty predictions and studying the interactions between these filters and images. This direction yields both an interpretable and intuitive understanding of model behaviors.

## 5 ETHICS STATEMENT

Although our formulation of parameter saliency is not restricted to image datasets and CNNs, we only conduct experiments in these settings. In contrast, real-world data and models come in many forms. Explainability methods which shed light in some settings may fail to do so in others. Moreover, we emphasize that some erroneous model behaviors are simply difficult to understand through existing methods, and the capabilities of parameter saliency are limited. In many applications, it is imperative that practitioners understand why their models behave as they do and that they are able to diagnose problems when they arise. We hope that our work helps to enable solutions to real-world problems. However, we caution against a false sense of security. Our visual interpretations of model behavior should be viewed as approximations since neural networks are incredibly complex.

## 6 REPRODUCIBILITY STATEMENT

We include our implementation of our parameter saliency method as well as the input-space saliency counterpart as a supplementary material. All the datasets used in the experiments are publicly available. The implementation details are available in Appendix B.

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

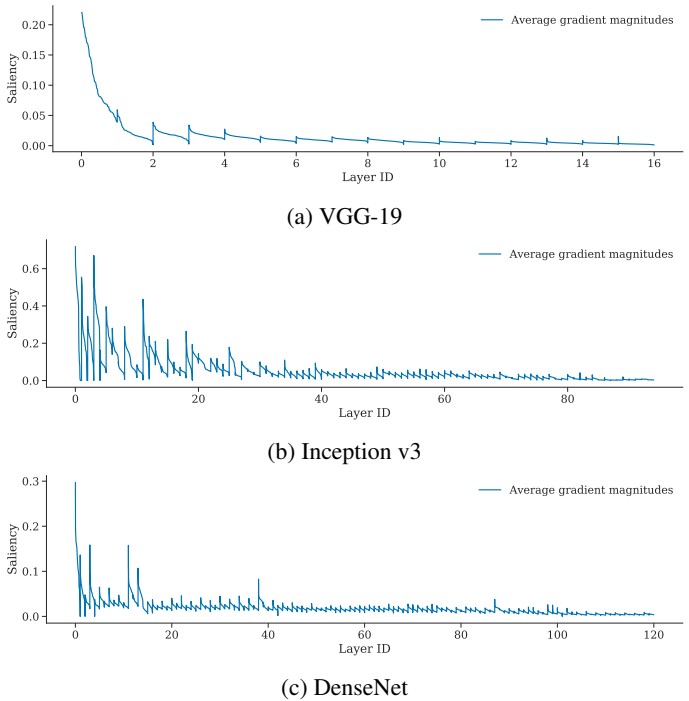

(a) VGG-19

(b) Inception v3

(c) DenseNet

Figure 9: **Filter-wise saliency profiles for other architectures.** (a) VGG-19 saliency profile (without standardization). (b) Inception v3 saliency profile (without standardization). (c) DenseNet saliency profiles (without standardization). In each panel the filter-wise saliency profile is averaged over the ImageNet validation set. In every panel, the filter saliency values in each layer are sorted in descending order, and each layer's saliency values are concatenated. The layers are displayed left-to-right from shallow to deep and have equal width on x-axis.

## A   ADDITIONAL EXPERIMENTS

### A.1   PARAMETER SALIENCY PROFILES FOR OTHER NETWORK ARCHITECTURES

In this section, we present average saliency profiles for several popular network architectures other than ResNet-50 (He et al., 2016). Analogously to Figure 1, Figure 9 presents average gradient magnitudes for VGG-19 (Simonyan & Zisserman, 2015), Inception v3 (Szegedy et al., 2016), and DenseNet (Huang et al., 2017). Similarly to Figure 2, we also present in Figure 10 standardized filter-wise saliency profiles for those architectures averaged across correctly and incorrectly classified ImageNet (Deng et al., 2009) samples.

### A.2   MORE EXAMPLES OF NEAREST NEIGHBORS

We present more examples of nearest neighbors in our parameter saliency space. Figure 12 are nearest neighbors in CIFAR-10 (Krizhevsky, 2009) dataset, where reference images are chosen from samples misclassified by our classifier. Figure 13 are examples from ImageNet, where images are captioned with the true label of the reference images.

In addition, in comparison with the nearest neighbor in our parameter saliency space, we also conduct the nearest neighbor search in the feature representation space. We take the feature representation from the `conv5_3` layer of a ResNet-50, and run the nearest neighbor search using the same reference images and candidate pool as in Figure 4. Results are shown in Figure 11. We find that nearest neighbors in the feature space bear more resemblance in image structures, but it fails to identify samples that share the same mistakes. In fact, most of the nearest neighbors in Figure 11 are correctly classified samples.

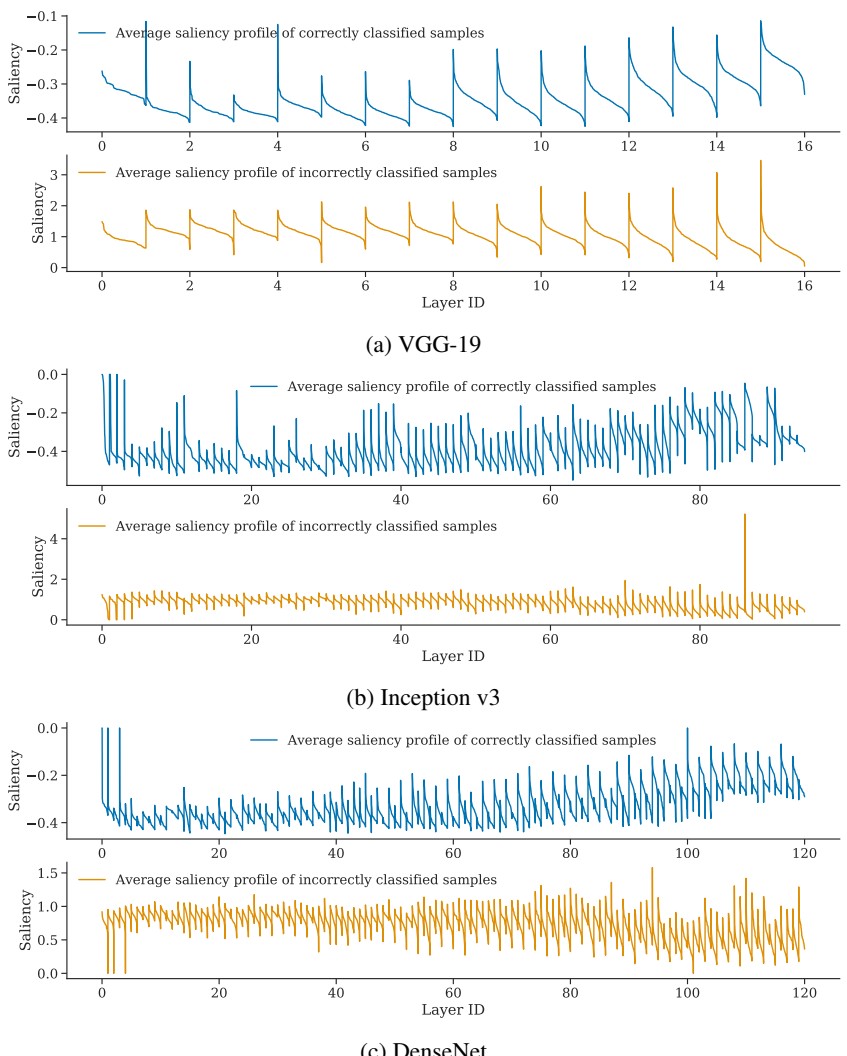

(a) VGG-19

(b) Inception v3

(c) DenseNet

Figure 10: **Standardized saliency profiles averaged over correctly vs incorrectly classified samples.** (a) VGG-19 saliency profiles. (b) Inception v3 saliency profiles. (c) DenseNet saliency profiles. In each panel, the top row presents the standardized saliency profiles averaged over correctly classified samples and the bottom row shows standardized saliency profiles averaged over incorrectly classified samples. On every panel, the filter saliency values in each layer are sorted in descending order, and each layer's saliency values are concatenated. The layers are displayed left-to-right from shallow to deep and have equal width on x-axis.

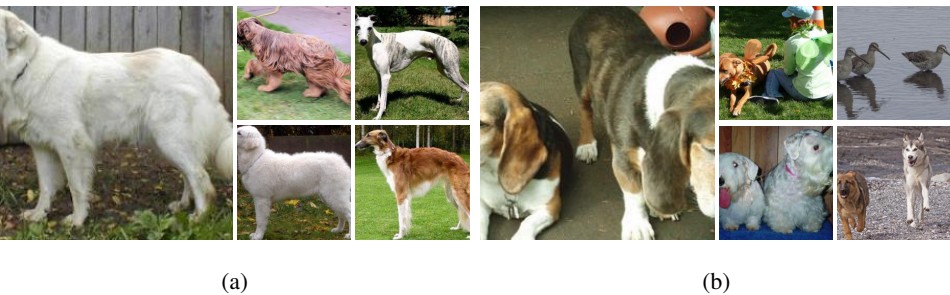

(a)                                                (b)

Figure 11: **Examples of nearest neighbors in the feature representation space (from ImageNet)**.

Furthermore, we present preliminary results on applying our method to language models. We use a BERT (Devlin et al., 2018) model, pre-trained on the task of predicting the word at a masked position. We consider an independent dataset for evaluation in our experiments, which consists of short sentences with masked words, provided by LAMA (F. Petroni & Riedel, 2019; Petroni et al., 2020), an open-source language model analysis framework. Similar to the filter-wise aggregation in convolutional networks, we adopt column-wise aggregation for obtaining our saliency profiles in the transformer-based architecture. We conduct the nearest neighbor search by comparing sentences from the dataset in saliency space and analyzing the top-5 nearest neighbors.

We present four examples below, where the numbering indicates $n$-th nearest neighbor sentence, and the italic word is the ground truth. Each example is accompanied with a description of similarities between the neighbors:

Reference: "Cany Ash and Robert Sakula are both Architects." incorrectly predicted as: actors

1. "David Castlles-Quintana and Vicente Royuela are *economists*." incorrectly predicted as: actors
2. "Raghuram Rajan is an *economist*." incorrectly predicted as: actor
3. "Richard G. Wilkinson and Kate Pickett are *British*." incorrectly predicted as: actors
4. "Nathan Alterman was a *poet*." correctly predicted: poet
5. "Zbigniew Badowski is an *architect*." incorrectly predicted as: author

Note that all 5 neighboring sentences here share the common structure of being declarations of profession for specifically named people. Interestingly, the first three closest sentences to the reference incorrectly predict the profession to be an actor as well. Moreover, the ground truth of the reference and its closest neighbor is economist/s.

Reference: "D'Olier Street is in Dublin." incorrectly predicted as: Paris

1. "A group who call themselves Huguenots lives in *Australia*." incorrectly predicted as: France
2. "Huguenots and Walloons settled in *Canterbury*." incorrectly predicted as: France
3. "In the Treaty of Lisbon 2007 *Ireland* refused to consent to changes." incorrectly predicted as: it
4. "Samuel Marsden Collegiate School is located in *Wellington*." incorrectly predicted as: Melbourne
5. "Konstantin Mereschkowski has *Russian* nationality." correctly predicted: Russian

Again, we see all five nearest neighbor sentences are semantically similar to the reference, this time relating to national affiliations and geography. In the first two closest sentences, the model incorrectly fills in a location with France, which is similar to the reference which incorrectly predicts Paris.

Reference: "The Super Bowl sponsor was the *Gap* clothing company." incorrectly predicted as Nike

1. "During Super Bowl 50 the *Nintendo* gaming company debuted their ad for the first time." incorrectly predicted as: video
2. "Experimental measurements on a model steam engine was made by *Watt*." incorrectly predicted as: Siemens
3. "ABC's programming strategy was criticized in May 1961 by *Life* magazine." incorrectly predicted as: Time
4. "In 2009, Doctor Who started to be shown on Canadian cable station *Space*." incorrectly predicted as: CBC
5. "To emphasize the 50th anniversary of the Super Bowl the *gold* color was used." incorrectly predicted as: blue

All but one of the nearest neighbors in this example relate to some kind of TV programming, and two also mention the Super Bowl. Much like the reference sentence, which incorrectly predicts the name of a corporation, the first four neighbors have a ground truth or incorrect prediction that is also a corporation.

Reference sample: "Tetzel's collections of money to free souls from purgatory was objected by *Luther*." incorrectly predicted as: some

1. "Newcastle was granted a new charter in 1589 by *Elizabeth*." incorrectly predicted as: Parliament

2. "The suggestion that imperialism was the "highest" form of capitalism is from *Lenin*." incorrectly predicted as: Aristotle

3. "Fritschel said the man's sleep was disturbed by *dreams*." incorrectly predicted as: lightning

4. "The concept that falling objects fell at the same speed regardless of weight was introduced by *Galileo*." incorrectly predicted as: NASA

5. "One of the earliest examples of Civil Disobedience was brought forward by the *Egyptians*." incorrectly predicted as: government

This is an example where there is a weak relation between the incorrect classifications (three of five involve some kind of government or government organization) of the nearest neighbors, but the sentences are still highly semantically similar. All of the neighbors except the third are declarations of historical actions performed by a specific person or group of people.

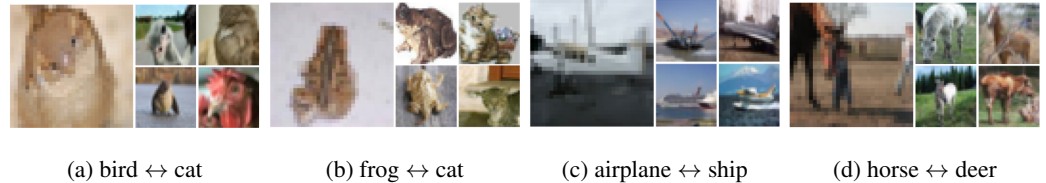

(a) bird ↔ cat          (b) frog ↔ cat          (c) airplane ↔ ship          (d) horse ↔ deer

Figure 12: **CIFAR-10 examples of nearest neighbors in parameter saliency space**. On CIFAR-10 images that cause similar filters to malfunction are often misclassified in a similar way.

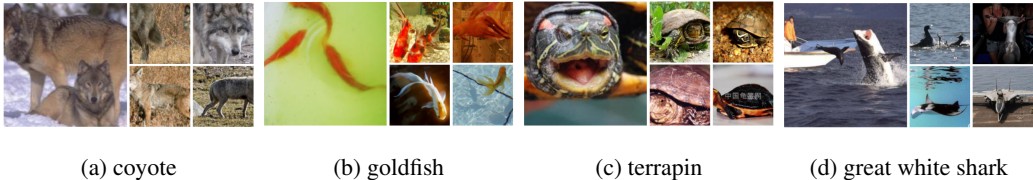

(a) coyote          (b) goldfish          (c) terrapin          (d) great white shark

Figure 13: **ImageNet examples of nearest neighbors in parameter saliency space**. In every panel, the reference image is in the left column and its nearest neighbors are in the right column. Panels are captioned by the true label of their reference image.

### A.3    CORRECTING MISTAKES ON OTHER DATASETS

We have shown in Section 3.3 that updating a few salient parameters of an ImageNet pre-trained network is enough for effectively correcting mistakes on the ImageNet validation set. Moreover, in this section, we show that this effect can be extended to other independent datasets. We use the ImageNet-v2 test set, consisting of 10,000 images collected by Recht et al. (2019), independent from the original ImageNet data.

In Figure 14, we observe the similar pattern as in Figure 6. With test samples independent of the training data, salient parameters still demonstrate the most strength in correcting mistakes and their nearest neighbors. The advantage is more prominent when inspecting model's predictions of unseen nearest neighbors.

### A.4    FINE-TUNING SALIENT FILTERS OF A VGG-19

In this section, we conduct the fine-tuning experiment introduced in Section 3.3 on a VGG-19 network trained on ImageNet, which has a total of 5504 filters. The learning rate for training our VGG network

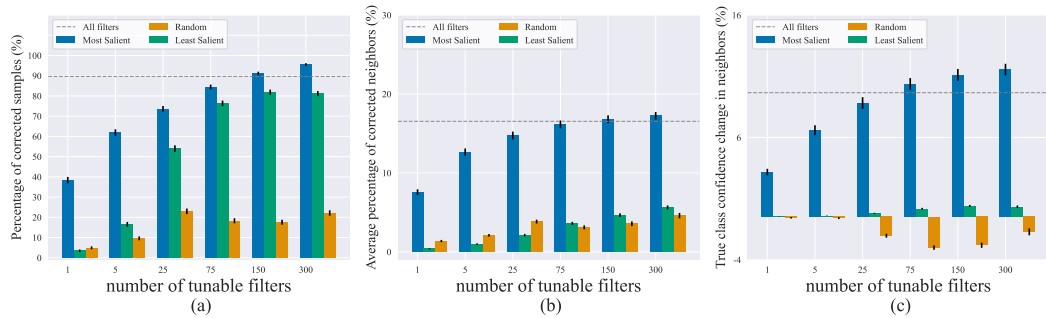

Figure 14: **Effect of updating a small number of filters on the ImageNet-v2 test data.** (a) Percentage of samples that are corrected after fine-tuning. (b) Average percentage of nearest neighbors that are also corrected after fine-tuning. (c) Average change in the confidence score of the true class among nearest neighbors. The horizontal line in each plot is the effect of updating the entire network.

is 1/10 of that for the ResNet, so we decrease the fine-tuning step size by 10 in this experiment. Figure 15 shows the effect of updating salient filters of a VGG-19. Note that we use the same range for the number of tunable filters in this experiment; 300 filters correspond to 5.5% of total filters in a VGG-19, while it is 1.2% for a ResNet-50.

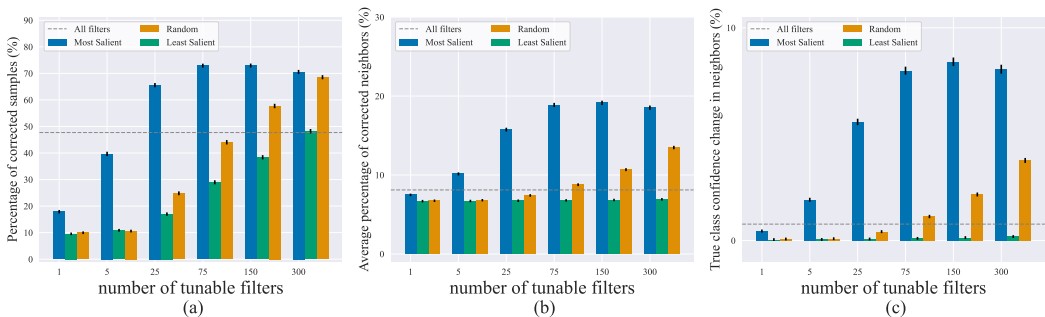

Figure 15: **Effect of updating a small number of filters on VGG-19.** (a) Percentage of samples that are corrected after fine-tuning. (b) Average percentage of nearest neighbors that are also corrected after fine-tuning. (c) Average change in the confidence score of the true class among nearest neighbors. The horizontal line in each plot is the effect of updating the entire network.

### A.5 RANDOM PERTURBATION OF SALIENT FILTERS

As an alternative to the pruning approach described in Section 3.1, random perturbations could be used to show that the most salient filters are indeed responsible for misclassification. We perturbed the filters using small Gaussian noise $\mathcal{N}(0, 0.001)$. Figure 16 presents the effect of randomly perturbing salient filters, we observe similar trends to our pruning experiments in Section 3.1.

### A.6 CONNECTION TO ADVERSARIAL ATTACKS IN PARAMETER SPACE

Adversarial attacks in parameter space have been used for optimizers which find flat loss minima (Foret et al., 2020; Kwon et al., 2021; Du et al., 2021) and for improving model robustness through parameter-corruption-resistant training (Sun et al., 2020).

One could instead apply adversarial attacks to construct parameter saliency profiles – choose a constraint space and perturb parameters in order to minimize loss, subject to the constraint, using the perturbation to parameters as a saliency profile (perhaps standardizing afterwards). We notice several advantages and disadvantages of this alternative. On the one hand, the "adversarial" approach requires a choice of constraint space and may require more compute (our method results in the exact

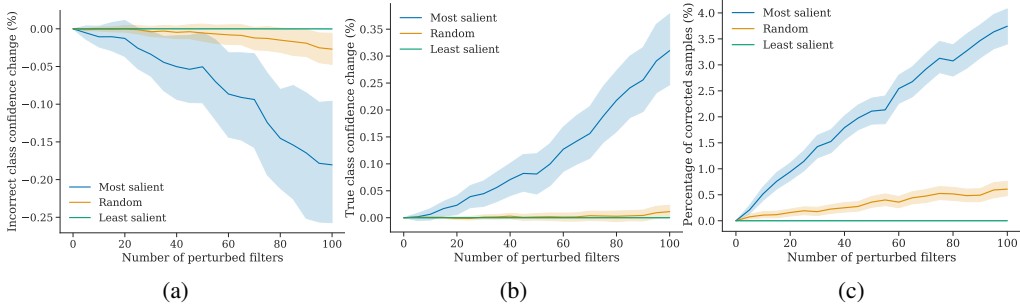

Figure 16: **Effect of randomly perturbing filters.** (a) Change in incorrect class confidence score. (b) Change in true class confidence score. (c) Percentage of samples that were corrected as the result of pruning filters. These trends are averaged across all images misclassified by ResNet-50 in the ImageNet validation set. The error bars represent 95% bootstrap confidence intervals.

same saliency profile as a single-step adversary). On the other hand, the choice of constraint space and optimizer in the adversarial approach yields more flexibility.

In this section, we compare our method with the adversarial-attack-based method. More specifically, for a given sample $(x, y)$, we perturb parameters either to maximize or minimize the loss subject to an L2-norm constraint on the parameter change:

$$\min_{\theta} / \max_{\theta} \mathcal{L}_{\theta}(x, y) \tag{4}$$
$$\text{s.t. } \|\theta - \theta_0\| < \epsilon$$

Then, given the adversarially perturbed parameters $\theta^*$, we define the adversarial-attack-based parameter saliency profile of the sample $(x, y)$ as the absolute difference from the initial parameters $\theta_0$: $s(x, y) = |\theta^* - \theta_0|$.

We compare the resulting adversarial saliency profiles with our original method on a random sample of 100 images from the ImageNet validation set. We observe that for a reasonably small constraint ($\epsilon = 10^{-4}$), the resulting adversarial saliency profiles are similar to our original parameter saliency profiles (as one would expect for smooth loss) with the average cosine similarity between the saliency profiles generated by each method for the same images reaching 0.99 (the average is taken over the random sample of 100 images). We also see that both methods agree on the top-k (we tried k=100) most salient filters: on average, 95% of filters identified by our original method as top-k salient filters were also identified as top-k salient filters by the adversarial parameter saliency. The differences were gradually more distinct with larger constraints, however, we note that the smaller epsilons are of greater interest since they reflect the intuition of perturbing only the most important parameters.

We also tried L1-regularized adversarial attacks. We can similarly enforce sparsity in our original method by simply adding the L1 regularizer to our loss before computing the gradient:

$$s(x, y)_i := |(1 - \alpha)\nabla_{\theta_i}(\mathcal{L}_{\theta}(x, y) + \alpha\|\theta - \theta_0\|_1)| \tag{5}$$

This modification can be seen as one step of the L1-regularized adversarial attack, and we experimentally checked that it produces very similar results with the cosine similarity between the resulting saliency profiles of 0.97 on average (for sufficiently large $\alpha = 0.99$).

### A.7 ADDITIONAL CASE STUDY FIGURES

**Masking non-salient parts of the image.** As noted in section 3.4 and presented in Figure 17, masking the non-salient parts of the image results in even worse misclassification confidence with the incorrect class confidence increasing compared to the reference image.

**Pasting the salient region from the reference image onto other "great white shark" images.** As mentioned in section 3.4, in order to further investigate the effect of the salient region, we pasted it (i.e., the seal and the boat) from the original image onto other images with "great white shark" ground truth label that were correctly classified by ResNet-50 (see Figure 18 for examples). We observed that this increased the probability of "killer whale" for 39 out of 40 examples of great white sharks from the ImageNet validation set with an average increase of 3.75%.

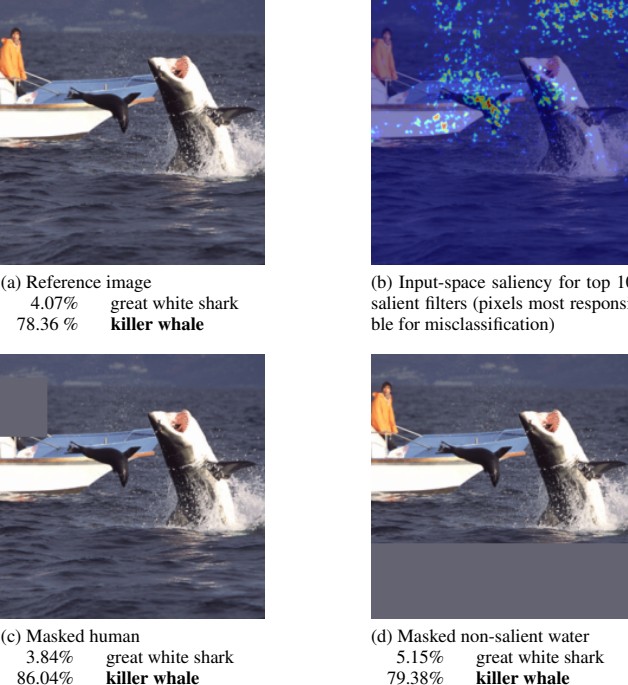

(a) Reference image
    4.07%      great white shark
    78.36 %    **killer whale**

(b) Input-space saliency for top 10 salient filters (pixels most responsible for misclassification)

(c) Masked human
    3.84%      great white shark
    86.04%    **killer whale**

(d) Masked non-salient water
    5.15%      great white shark
    79.38%    **killer whale**

Figure 17: **Masking non-salient parts of the image.** (a) Reference image of "great white shark" misclassified by the model as "killer whale" and the corresponding confidence scores. (b) Pixels that cause the top 10 most salient filters to have high saliency. (c) Masked (non-salient) human. (d) Masked non-salient water region.

**Examples of images with "killer whale" label.** As we discussed in section 3.4, the model might have learned to correlate a combination of sea creatures (e.g. a seal) and man-made structures (e.g. a boat) with the "killer whale" label. Images of killer whales in ImageNet often have man-made structures which look similar to the boat, we provide examples of that in Figure 19.

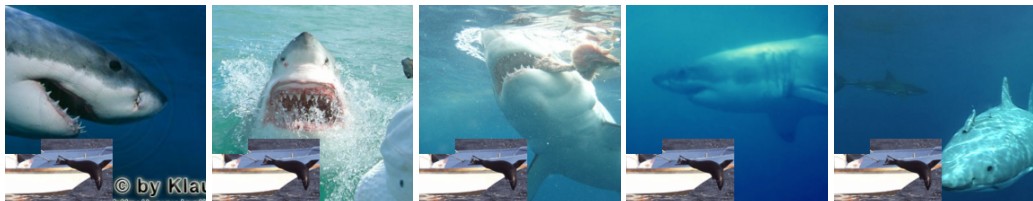

Figure 18: **Sample "great white shark" images with boat and seal.** The salient region from the case study image pasted onto other "great white shark" images.

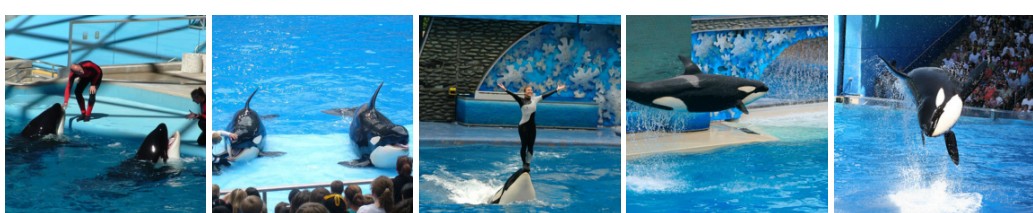

Figure 19: **ImageNet examples of "killer whale".**

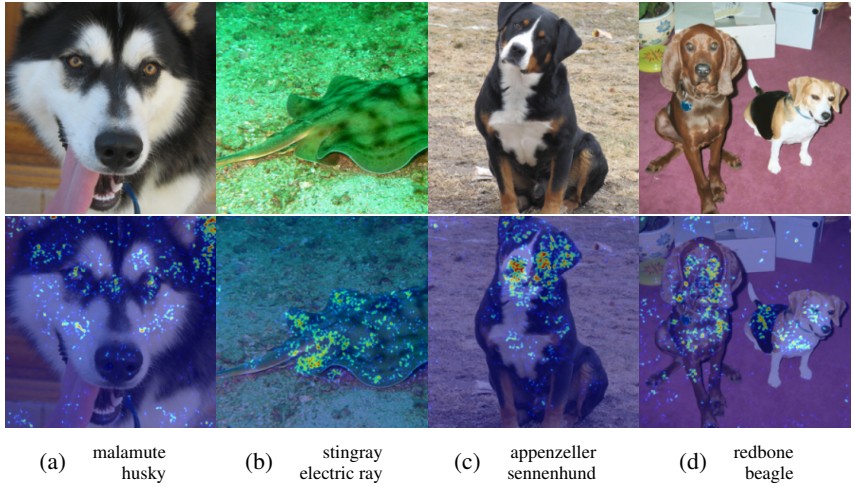

| (a) | malamute
husky | (b) | stingray
electric ray | (c) | appenzeller
sennenhund | (d) | redbone
beagle |
|-----|-------|-----|-------|-----|-------|-----|-------|

Figure 20: **Pixels responsible for mistakes focused on the target object.** (a)-(b) Masking the salient pixels corrects the misclassification where masking confusing features (e.g. dog ears or spot patterns) helps distinguish animals. (c)-(d) Masking the salient pixels results in correct class confidence decrease, when the salient pixels are densely focused on the target object. The correct class label is specified in the top row and the predicted incorrect class label – in the bottom row of the subcaption on each panel.

## A.8    EXPLORING NEURAL NETWORK MISTAKES

In Section 3.4, we apply our parameter-space saliency method as an explainability tool using it alongside our input-space technique to study how image features cause specific filters to malfunction. In our case study, the salient pixels that confuse the network are focused less on the target object than on other image features, and masking the salient regions which are not on the target object improves the network behavior. Such examples expose the network's reliance on spurious correlations and constitute an interesting type of model mistake.

The masking approach can be adopted to explore network mistakes and find other interesting cases (e.g., cases where the salient pixels are not concentrated on the target object). Investigating examples where masking the most salient pixels improves performance may provide insights into the model's misbehavior as well as expose dataset noise and biases. We select misclassified samples where masking the top 5% of salient pixels leads to an increase of at least 25% in confidence corresponding to the correct class. We showcase representative examples of different types of mistakes that we observe in those samples in Figure 8. Many of the neural network misclassifications stem from classifying a non-target object in images with multiple objects (see Figure 8 (a), (b)). However, other mistakes are triggered by background features (Figure 8 (d)), dataset biases (as our case study experiments in Figure 7 suggest), and label noise (Figure 8 (e)).

Interestingly, in some of the selected cases the salient regions still focus on the target object features (see Figure 8 (c)), and masking them improves the model's behavior. Masking salient target object features that confuse the network seems to be particularly beneficial for images of animals; for example, masking dog ears helps the network identify the correct breed (see Figure 20(a)) or masking spot patterns helps distinguish different types of rays (see Figure 20(b)).

While masking the top 5 % of salient pixels results in correct classification for all samples in Figure 8 and in Figure 20(a), (b), this is, of course, not always the case. Sometimes, pixels which cause large filter saliency values are focused densely on the target object, and masking them results in decreased confidence corresponding to the correct class. Selecting such samples can be used to find situations where the network is genuinely confused by the target object rather than background features (Figure 20 (c)) or samples with multiple objects present and with salient pixels more focused on the target object (Figure 20 (d)).

## A.9 COMPARISON TO GRADCAM

Existing input saliency maps used with the predicted label can highlight features which are related to misclassification. However, they are not specifically geared towards that goal. Our input saliency technique highlights image features that cause specific filters to malfunction and those features, while they might in some cases coincide with the features that explain a high class confidence score corresponding to the predicted label, may not be the same.

In this section, we will compare our input-space saliency technique which highlights pixels that drive high parameter saliency values of specific filters (i.e., pixels that confuse the network) to the visual explanations produced by GradCAM with the predicted label [3] (Selvaraju et al., 2017) – one of the most popular and high quality input-saliency methods.

Figures 21 and 22 present panels of images comparing our method to GradCAM explanations computed with the predicted label. From the perspective of highlighting pixels responsible for neural network mistakes and for driving high filter saliency values, we note the following:

- GradCAM highlights the object that corresponds to the incorrect label, and the entire target object is highlighted in the images where only the target object is present (see Figure 21 (c)-(e), Figure 22 (a)-(c)). However, when our method focuses on the target object, it highlights specific features of that object. Those are the features that confuse the network, and masking them can correct the misclassification.

- In cases where the network classifies the non-target object in the image (see Figure 21 (a)-(b), Figure 22 (d)), both methods highlight the non-target objects. However, GradCAM is more localized to the non-target object. This is expected since GradCAM produces visual explanations for the predicted label (and has been shown to produce highly localized saliency maps (Selvaraju et al., 2017)) while our method highlights all pixels that drive the filter saliency, and these pixels may be located on the target object as well.

- In cases where the misclassification does not stem from confusion by the target object or classifying the non-target object (see Figure 21 (d)-(e), Figure 22 (e)), our method highlights background features and/or a combination of non-target object features, while GradCAM still highlights the target object. For example, in Figure 22 (e), our method highlights the boat and the sky much more than GradCAM, and our case study masking experiments in section 3.4 show that those regions indeed confuse the network.

- In addition, we emphasize that our input saliency technique is specific to the chosen filter set $F$ and is introduced to study the interaction between the image features and the malfunctioning filters. In contrast, GradCAM is not able to relate image-space mistakes to an arbitrary set of model parameters or filters chosen by the user.

To summarize, GradCAM (as well as many other input-space saliency methods) was designed to be highly localized to the object correponding to the label of interest, while our method highlights sparse fine-grained features of images which we believe is a desirable property for our specific application. Therefore, we opt for using input-gradient information similar to the original Vanilla Gradient (Simonyan et al., 2014) method. However, instead of class confidence scores, we use a different loss – cosine distance to the boosted parameter-saliency profile (as described in Section 2.2) which allows us to explore how image features cause specific filters to malfunction.

## A.10 INPUT-SALIENCY SANITY CHECK

To assure that our input-space saliency technique is model dependent, we performed the model randomization test from (Adebayo et al., 2018). We can see that the input saliency map is model dependent. We note that the data randomization test is not applicable in our case because our input-space saliency map is based on the parameter-saliency profile and parameter-saliency is designed to investigate a pretrained model with particular weights.

---

[3]Implementation from https://github.com/kazuto1011/grad-cam-pytorch under MIT license

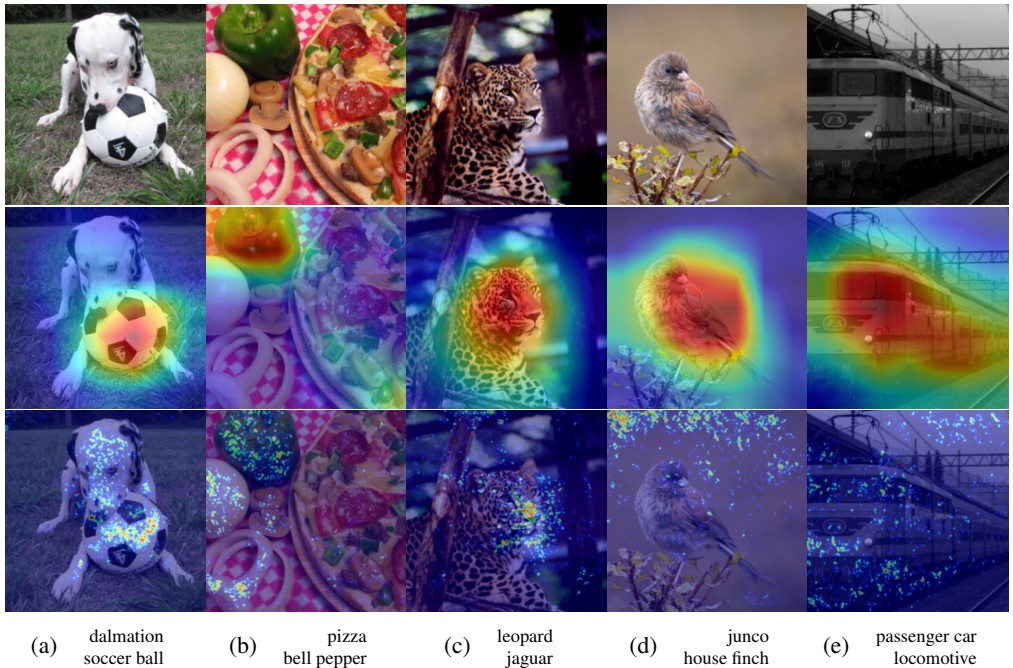

(a) dalmation / soccer ball  (b) pizza / bell pepper  (c) leopard / jaguar  (d) junco / house finch  (e) passenger car / locomotive

Figure 21: **Comparison to GradCAM.** Top row: original image. Middle row: GradCAM input-space saliency map for the predicted label. Bottom row: our input-space saliency technique which highlights pixels that drive high parameter saliency values of specific filters (i.e., pixels that confuse the network). The correct class label is specified in the top row and the predicted incorrect class label – in the bottom row of the subcaption on each panel.

# B    Implementation details

## B.1    Hyper-parameter setting for fine-tuning salient filters

When tuning a small number of random or least salient filters, we re-normalize the gradient magnitude of these parameters to be the same as the salient filters for a fair comparison; otherwise the gradients for these parameters are always smaller than the salient ones by the definition of our saliency profile, and updating them would make less change to a model than updating the salient ones. In addition to re-normalizing the gradients, we also multiply them with a step size, similar to the concept of learning rate in stochastic gradient descent. For ResNet-50, we set the step size to be 0.001, which equals to the learning rate of the last epoch when training the ResNet-50 on ImageNet from scratch. For VGG-19, we also set the fine-tuning step size to be the learning rate from the last training epoch, which is 0.0001. We also note that the batch normalization layers were set to the test mode for our fine-tuning experiments.

## B.2    Input saliency visualization

The number of top salient filters to boost was chosen to be 10 in all input-space saliency experiments. The filters were boosted by multiplying by 100. For visualization, absolute input gradients were thresholded at 90-th percentile and Gaussian Blur with $(3, 3)$ kernel was applied.

## B.3    Hardware

The experiments were run on Nvidia GeForce RTX 2080Ti GPUs with 11Gb GPU memory on a machine with 4 cpu cores and 64Gb RAM. The input-space and parameter-saliency profiles take seconds to compute for a single sample.

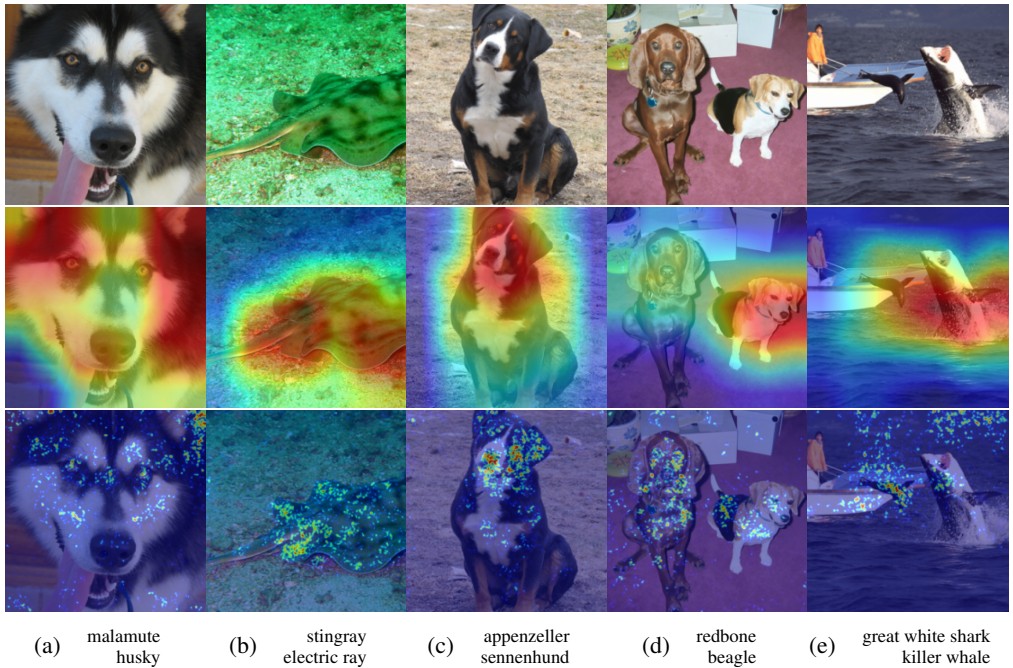

| (a) | malamute
husky | (b) | stingray
electric ray | (c) | appenzeller
sennenhund | (d) | redbone
beagle | (e) | great white shark
killer whale |

Figure 22: **Comparison to GradCAM.** Top row: original image. Middle row: GradCAM input-space saliency map for the predicted label. Bottom row: our input-space saliency technique which highlights pixels that drive high parameter saliency values of specific filters (i.e., pixels that confuse the network). The correct class label is specified in the top row and the predicted incorrect class label – in the bottom row of the subcaption on each panel.

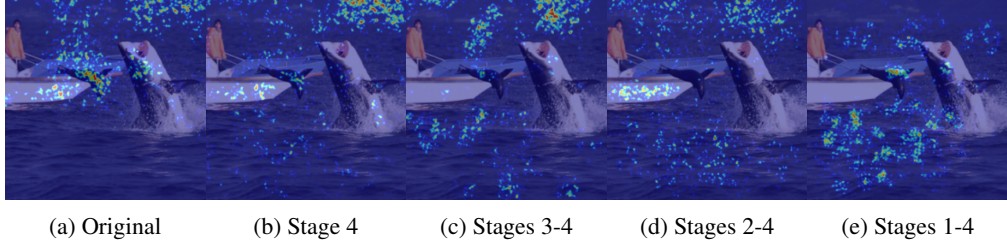

(a) Original      (b) Stage 4      (c) Stages 3-4      (d) Stages 2-4      (e) Stages 1-4

Figure 23: **Sanity checks.** (a) No randomization of ResNet-50. (b) Only stage 4 of ResNet-50 is randomized. (c) Stages 3-4 of ResNet-50 are randomized. (d) Stages 2-4 of ResNet-50 are randomized. (e) The entire ResNet-50 is randomized.

