# OpenReview forum: "Where do Models go Wrong? Parameter-Space Saliency Maps for Explainability"
_ICLR.cc/2022/Conference — ICLR 2022 Submitted_

### Official Review · Reviewer_sCeW · 2021-10-21

**Correctness:** 4
**Technical Novelty And Significance:** 3
**Empirical Novelty And Significance:** 3
**Recommendation:** 5
**Confidence:** 5

**Main Review:**

[1] Eq. 1 is not clear enough, what's the meanings of K.

[2] If the proposed analytic method only suits for the image classification task or the test models, the contribution would be limited.

[3] As stated in Section 3.3, we can correct mistakes by fine-tuning salient filters. Does it mean a model of general performance could become a top-performing model benefiting from the proposed fine-tuning method?

[4] Whether the performance-gain obtained by fine-tuning salient filters can transfer to the independent database, rather than the closed test set.

**Summary Of The Paper:**

This paper devises an analytic method for explainability based on the observation of filter-wise parameter saliency distribution, and tests on several models. And several experiments are conducted to deminstrate the conjecture. The motivation is straightforward and easy to understand.

**Summary Of The Review:**

The contribution might be not enough for iclr.

---

> ### Author Response · Authors · 2021-11-22
> **Response to Reviewer sCeW**
>
> Dear Reviewer sCeW,
>
> Thank you for your feedback.
>
> [1] In these equations, k indexes over the set of filters in the network. For each filter, we have an indexing set $\alpha_k$, which points to individual parameters corresponding to the k-th filter. Thank you for pointing out this confusion, we have clarified it in our draft.
>
>
> [2] Prompted by your review, we have now performed experiments on applying our parameter saliency method to language models. We use a BERT model, pre-trained on the task of predicting the word at a masked position. We consider an independent dataset for evaluation in our experiments, which consists of short sentences with masked words, provided by LAMA (https://github.com/facebookresearch/LAMA), an open-source language model analysis framework. Similar to the filter-wise aggregation in convolutional networks, we adopt column-wise aggregation for obtaining our saliency profiles in the transformer-based architecture. We conduct the nearest neighbor search by comparing sentences in saliency space, and for each example, we analyze the top-5 nearest neighbors.
>
> We present an example below, where the numbering indicates n-th nearest neighbor sentence in parameter saliency space, and the italic word is the ground truth:
>
> * Reference: “Cany Ash and Robert Sakula are both *Architects*.” incorrectly predicted as: **actors**
>
> 1. “David Castlles-Quintana and Vicente Royuela are *economists*.” incorrectly predicted as: **actors**
> 2. “Raghuram Rajan is an *economist*.” incorrectly predicted as: **actor**
> 3. “Richard G. Wilkinson and Kate Pickett are *British*.” incorrectly predicted as: **actors**
> 4. “Nathan Alterman was a *poet*.” correctly predicted: **poet**
> 5. “Zbigniew Badowski is an *architect*.” incorrectly predicted as: **author**
>
> We note that all 5 neighboring sentences  declare the profession of specific people. Moreover, the first three closest sentences also incorrectly predict the profession to be an actor.  Another example from the same dataset is the following:
>
> * Reference: “D'Olier Street is in *Dublin*.” incorrectly predicted as: **Paris**
>
> 1. “A group who call themselves Huguenots lives in *Australia*.” incorrectly predicted as: **France**
> 2. “Huguenots and Walloons settled in *Canterbury*.” incorrectly predicted as: **France**
> 3. “In the Treaty of Lisbon 2007 *Ireland* refused to consent to changes.” incorrectly predicted as: **it**
> 4. “Samuel Marsden Collegiate School is located in *Wellington*.” incorrectly predicted as: **Melbourne**
> 5. “Konstantin Mereschkowski has *Russian* nationality.” correctly predicted: **Russian**
>
> Again, we see all five nearest neighbor sentences are semantically similar to the reference, this time relating to national affiliations and geography. Moreover, the first two closest sentences involve cases where the model has also incorrectly filled in a location with something French.
>
> We thank the reviewer for their suggestion and we added these NLP experiments, which validate the utility of our method beyond image classification, to Appendix A.2.
>
> [3] We want to stress that while our saliency method can identify parameters which are useful for correcting mistakes without strongly affecting performance on other samples, fine-tuning salient parameters can have a non-zero impact on performance on other samples.  As we indicate in our paper, these experiments are intended to validate that the salient parameters we identify are indeed particularly important for predictions on similar samples, not as a state-of-the-art fine-tuning method.   In our experience, fine-tuning salient filters corrects predictions on the samples of interest but the overall test accuracy typically stays roughly the same.  However, performance on other samples degrades significantly less by fine-tuning salient parameters than by fine-tuning random ones.
>
>
>
> [4] In order to address your question concerning if fine-tuning transfers outside of the closed test set, we conduct additional experiments on ImageNet-V2 (Recht et al. 2019), a test set consisting of 10,000 images collected independently from the original ImageNet data. The new results suggest that the effect of updating salient filters transfers well to samples outside of the closed ImageNet test set and matches trends we previously observed in Figure 6.  The effect of updating salient filters on this ImageNet-V2 is included in appendix A.3 of our updated draft.
>
>
> Thank you for your comments. Do you have any additional comments or concerns we can address?

---

> > ### Author Response · Authors · 2021-11-29
> > **Thank you for your review**
> >
> > Dear Reviewer sCeW,
> >
> > Thank you again for your feedback and suggestions on improving our paper. We hope that our clarifications and additional experiments on extending our fine-tuning experiments to independent datasets and applying our parameter saliency method to language models address your concerns. We would very much appreciate it if you considered increasing the score of our paper.
> >
> > Thank you!

---

### Official Review · Reviewer_3o7Z · 2021-10-26

**Correctness:** 3
**Technical Novelty And Significance:** 3
**Empirical Novelty And Significance:** 3
**Recommendation:** 6
**Confidence:** 4

**Details Of Ethics Concerns:**

No concern.

**Main Review:**

+ Though the idea of parameter-wise visualization is not new, and widely studied in other works, this paper provides an interesting perspective on the parameters responsible for misclassification.
+ Meanwhile, the authors provides the thorough analysis and potential for applying the salient parameters on improving classification accuracy. This makes the paper with pratical use.

- The proposed idea is somehow similar to the parameter-space adversarial attacks. What if use attack methods to filter out the salient parameters that mislead the final classfications? This seems more intuitive.
- For the experiments of parameters pruning, it may be not reasonable for directly turn off the parameters, what if give them a perturbation instead? It can still reveal the sensitivity of the parameters.

**Summary Of The Paper:**

This paper introduces a parameter-space saliency map to explore the salient parameters that are responsible for miscalssification. A set of experiments and visulizations are conducted on the salient parameters, leading to several interesting findings, such as, the nearest parameter neighbors share similar semantic information. Besides, the authors are also trying to improve the prediction accuarcy by turning off or fine-tuning the salient parameters.

**Summary Of The Review:**

The proposed method is new as a parameter-wise visualization, but it becomes somehow similar to the  parameter-space adversarial attacks. The authors would better claim the similarity and difference of these 2 topics. Besides, adversarial attack needs to be reviewed as the related work.

---

> ### Author Response · Authors · 2021-11-22
> **Response to Reviewer 3o7Z**
>
> Dear Reviewer 3o7Z,
>
> Thank you for your positive feedback.
>
> We agree that there is a connection between our method and parameter-space adversarial attacks, thank you for pointing this out. One could instead choose a constraint space and perturb parameters in order to minimize loss, subject to the constraint, using the perturbation to parameters as a saliency profile (perhaps standardizing afterwards).  We notice several advantages and disadvantages of this alternative.  On the one hand, the “adversarial” approach requires a choice of constraint space and may require more compute (our method results in the exact same saliency profile as a single-step adversary).  On the other hand, the choice of constraint space in the “adversarial” approach yields more flexibility.
>
> Prompted by your review, we investigated this approach and conducted experiments to compare our method with adversarial attacks. More specifically, we perturb parameters either to maximize or minimize loss subject to an L2-norm constraint on parameter change.  Then, given the adversarially perturbed parameters $\theta^*$, we define the adversarial-attack-based parameter saliency profile of a sample $(x, y)$ as the absolute difference from the initial parameters $\theta_0$: $s(x,y) = |\theta^* - \theta_0|$.
>
> We compare the resulting adversarial saliency profiles with our original method on a random sample of 100 images. We observe that for a reasonably small constraint the resulting adversarial saliency profiles are similar to our original parameter saliency profiles (as one would expect for smooth loss) with the average cosine similarity between the saliency profiles generated by each method for the same images reaching 0.99 (the average is taken over the random sample of 100 images). We also see that both methods agree on the top-k (we tried k=100) most salient filters: on average, 95% of filters identified by our original method as top-k salient filters were also identified as top-k salient filters by the adversarial parameter saliency. The differences were gradually more distinct with larger constraints.
>
> We also tried L1-regularized adversarial attacks. We can similarly enforce sparsity in our original method by simply adding the L1 regularizer to our loss before computing the gradient.  This modification can be seen as one step of the L1-regularized adversarial attack, and we experimentally checked that it produces very similar results with the cosine similarity between the resulting saliency profiles of 0.97 on average.
>
> We have now added a section on the connection to adversarial attacks to Appendix A.6 of our draft and cited the relevant papers. Thank you again for this suggestion.
>
> Thank you for suggesting an alternative to the parameter pruning experiment. We performed the experiment with random perturbations, and it confirmed that the most salient parameters are indeed the ones most responsible for misclassification -- we saw similar trends to our pruning experiments (Figure 3). We added a figure with the results of this experiment to the Appendix A.5 of our current draft.

---

> > ### Author Response · Authors · 2021-11-29
> > **Thank you for your review**
> >
> > Dear Reviewer 3o7Z,
> >
> > Thank you again for your feedback and suggestions on improving our paper. We hope that our clarifications and additional experiments on comparison to adversarial attacks in parameter space and the random perturbation experiment address your concerns. We would very much appreciate it if you considered increasing the score of our paper.
> >
> > Thank you!

---

### Official Review · Reviewer_qR4u · 2021-11-03

**Correctness:** 4
**Technical Novelty And Significance:** 4
**Empirical Novelty And Significance:** 3
**Recommendation:** 6
**Confidence:** 2

**Main Review:**

Strengths
1. Different from the previous papers focusing on the saliency maps, this paper proposes to analyze the network parameters, which is novel and interesting.
2. Quantitative and qualitative experiments are conducted to verify the proposed standpoints.
3. The discoveries in the paper is interesting and inspired.


**Summary Of The Paper:**

This paper conducts some basic experiments based on the network parameters which are responsible for erroneous decisions. To support the proposed standpoints, the authors conduct a lot of quantitative and qualitative experiments.

**Summary Of The Review:**


I am not an expert in this field and I did read carefully about this paper. I could not find any weakness in this paper. The contributions proposed in this paper are really interesting and are verified by the proposed experiments.

---

> ### Author Response · Authors · 2021-11-22
> **Response to Reviewer qR4u**
>
> Dear Reviewer qR4u,
>
> Thank you very much for your positive feedback. We are encouraged that you found our paper novel and interesting!

---

> > ### Author Response · Authors · 2021-11-30
> > **Thank you for your review**
> >
> > Dear Reviewer qR4u,
> >
> > Thank you again for taking the time to review our paper and for your positive comments!

---

> > > ### Comment · Reviewer_qR4u · 2021-11-30
> > > **Modification about my rating**
> > >
> > > I have read all the reviews carefully and agree with Reviewer tkQp. So I downgrade my rating.

---

### Official Review · Reviewer_tkQp · 2021-11-16

**Correctness:** 2
**Technical Novelty And Significance:** 3
**Empirical Novelty And Significance:** 2
**Recommendation:** 5
**Confidence:** 4

**Main Review:**

The paper devises an approach so look at saliency of filters by aggregating the gradients w.r.t. the parameters belonging to that filter. In that sense it is an extension of sensitivity analysis. The paper then shows that only a limited number of filters need updating to modify the prediction. Given that some of these filters might have very many parameters the resulting change can be quite big.

I believe there are 2 clear limitations in the paper.
- it is not clear how this differs from an adversarial attack in parameter space/simply exploiting the sensitivity to certain parameters. I.e. changing parts with large gradients will change the behavior. The fact that this is done for a limited number of parameters is similar to using L1 normed adversarial attacks.
- The evaluation in section 3.4 is quite limited in scope and looks to be more anecdotal. Extending this to an in depth evaluation would improve the paper.


Detailed comments below


# Intro/Methods
- Figure 1/2: . I believe this figure can be improved by assigning each layer space corresponding of "unit 1" on the x-axis. Right now, the decay for the earlier layers (with fewer filters) is hard to see. By normalizing the space used by a single layer this might become easier to read.

- Figure 1 and 2: I am very surprised that in Figure 1 and Figure 2-bottom the saliency averaged over filters is always positive. In contrast it is always negative in Figure 2 top.  Fig 1: states that it is not standardized. Fig 2. does not specify this but it seems to me that there was an inconsistency.

- Section 2.2/ Eq 3  I did not find later on in the manuscript how the magnitude of the selected filters was increased. Instead of using the cosine distance, one could also try to only pass the gradients through the filters that were deemed most salient. But this might result in a very sparse heatmap.

# Experiments
- Section 3.1/Figure 3:  If I understand it correctly these plots only contain data from initially misclassified samples. While the evaluation is done over all misclassifications, it is the sum of experiments on single images. I.e. when a filter is removed this is done only for a single example and evaluated on this example. In that sense it is similar to an adversarial perturbation in the middle of the network. The fact that this changes the confidence should not be surprising per se.

- Section 3.2/Figure 4/5: The nearest neighbor approach is done by computing the cosine similarity of the saliency maps. Since this is dominated by the higher layers it should not be surprising that this results in conceptual similarity vs input similarity.  This can also explain why the same classes are misclassified since higher level concepts are also higher in the layers. In Fig. 5 similarity of low vs higher level layers is shown. The texts states that this is done to show misbehavior, but it is not clear what is misbehaving in this plot.


- Section 3.3 Correcting mistakes by updating salient filters is evaluated on all images independently. How was batch norm used in this case for the update? Setting it to training mode could have weird effects.  I am not convinced that it should be surprising that updating the ones with large gradients results in changes at the output.

- Section 3.4
This section is the most problematic to me since it lacks large scale evaluation and is mostly anecdotal.

Figure 8 is a set of
# General
Make the labels of the axes in plots larger. They are hard to read.

**Summary Of The Paper:**

The paper tries to identify the most salient parameters in a neural network and shows that when these parameters are modified the predictions change. They also show that there is a relation between these parameters and regions in the input that impact the classification result.

**Summary Of The Review:**

I tend to vote for rejection of this paper. The effects described are similar to adversarial attacks and in most but not all cases the influence of a parameter is only investigated on the image where the saliency was computed.
The evaluation of the proposed approach in terms of providing pixel wise interpretability section 3.4 is limited.

This review was done after the rebuttal period as an emergency review. I had no access to the discussion with the authors which might impact my understanding.

---

> ### Author Response · Authors · 2021-11-22
> **Part 1: Addressing the two main concerns of Reviewer tkQp**
>
> Dear Reviewer tkQp,
>
> Thank you for your thoughtful feedback.
>
> * We agree that there is a connection between our method and parameter-space adversarial attacks, thank you for pointing this out. One could instead choose a constraint space and perturb parameters in order to minimize (or maximize) loss, subject to the constraint, using the perturbation to parameters as a saliency profile (perhaps standardizing afterwards).  We notice several advantages and disadvantages of this alternative.  On the one hand, the “adversarial” approach requires a choice of constraint space and may require more compute (our method results in the exact same saliency profile as a single-step adversary).  On the other hand, the choice of constraint space in the “adversarial” approach yields more flexibility. \
> \
>  We investigated this approach and conducted experiments to compare our method with adversarial attacks. More specifically, we perturb parameters either to maximize or minimize loss subject to an L2-norm constraint on parameter change.  Then, given the adversarially perturbed parameters $\theta^*$, we define the adversarial-attack-based parameter saliency profile of a sample $(x, y)$ as the absolute difference from the initial parameters $\theta_0$: $s(x,y) = |\theta^* - \theta_0|$.\
> \
>  We compare the resulting adversarial saliency profiles with our original method on a random sample of 100 images. We observe that for a reasonably small constraint the resulting adversarial saliency profiles are similar to our original parameter saliency profiles (as one would expect for smooth loss) with the average cosine similarity between the saliency profiles generated by each method for the same images reaching 0.99 (the average is taken over the random sample of 100 images). We also see that both methods agree on the top-k (we tried k=100) most salient filters: on average, 95% of filters identified by our original method as top-k salient filters were also identified as top-k salient filters by the adversarial parameter saliency. The differences were gradually more distinct with larger constraints.\
>  We also tried L1-regularized adversarial attacks. We can similarly enforce sparsity in our original method by simply adding the L1 regularizer to our loss before computing the gradient.  This modification can be seen as one step of the L1-regularized adversarial attack, and we experimentally checked that it produces very similar results with the cosine similarity between the resulting saliency profiles of 0.97 on average.\
>  We have now added a section on the connection to adversarial attacks to Appendix A.6 of our draft and cited the relevant papers.
> \
> \
> While we are aware of adversarial attacks in parameter space used for optimizers which find flat loss minima and are used for improving model robustness (e.g., [1]), to the best of our knowledge, adversarial attacks in parameter space have not been used for the explainability purposes we study in this work.
>
>
> * We want to point out that our case study in Section 3.4 was intended as an example of how our method could be used to explore neural network mistakes. We would also like to add that we do dataset-level quantitative validations of our parameter saliency method earlier in the paper, in sections 3.1 and 3.3. \
> \
> We agree, however, that a large-scale version of the case study experiment could provide additional insights. In fact, we tried automating that experiment by masking out the pixels most responsible for high filter saliency values on the dataset level, but we can only mask out pixels in the background in order to avoid erasing the foreground object.  Therefore, performing this experiment on the dataset level requires a classification dataset with additional background masks so that we can ensure that we only mask out salient pixels in the background, and such annotations were not available for ImageNet. Prompted by your review, we found a suitable dataset with background masks, Causal ImageNet, which is a concurrent work and is currently under review for ICLR 2022 (https://openreview.net/forum?id=XVPqLyNxSyh). We are now running a dataset level version of the case study experiment using the core object annotations generated by the Causal ImageNet approach. We note that we are still working on these experiments as this review was only recently posted, and we will include the results in the camera-ready version of our paper. Thank you for suggesting this direction!
>
> References\
> [1] Foret, Pierre, et al. "Sharpness-aware minimization for efficiently improving generalization.", ICLR 2021

---

> ### Author Response · Authors · 2021-11-22
> **Part 2: Addressing the detailed comments on Intro/Methods**
>
> **Intro/Methods**
>
> * *"Figure 1/2: I believe this figure can be improved by assigning each layer space corresponding of "unit 1" on the x-axis. Right now, the decay for the earlier layers (with fewer filters) is hard to see. By normalizing the space used by a single layer this might become easier to read."* \
> \
> Thank you, we incorporated your suggestion and improved our visualizations.
>
>
> * *“Figure 1 and 2: I am very surprised that in Figure 1 and Figure 2-bottom the saliency averaged over filters is always positive. In contrast it is always negative in Figure 2 top. Fig 1: states that it is not standardized. Fig 2. does not specify this but it seems to me that there was an inconsistency.”*  \
> \
> Figure 2 presents standardized saliency profiles in both panels (we mention that the profiles are standardized in the name of the figure). Thank you for pointing out the confusion, we further clarified the caption by replacing “saliency profile” with “standardized saliency profile”. \
> \
> The idea of Figure 2 is to show that incorrectly classified examples are more salient on average than correctly classified examples. The top panel of Figure 2 presents the average standardized saliency profile of correctly classified samples, and the bottom panel of Figure 2 presents the average standardized saliency profile of incorrectly classified samples. The standardized saliency profiles were computed for each sample (the standardization was done with the respect to the entire ImageNet validation set) before taking the average over correctly/incorrectly classified samples to produce the plots.
>
>
> * *“Section 2.2/ Eq 3 I did not find later on in the manuscript how the magnitude of the selected filters was increased. Instead of using the cosine distance, one could also try to only pass the gradients through the filters that were deemed most salient. But this might result in a very sparse heatmap.”* \
> \
>  In the paragraph before Equation (3), we mention that a boosted saliency profile $s'_F$ is created by increasing the entries of $\hat{s}$ corresponding to the chosen filters $F$ by multiplying them by a large constant. In our experiments, we multiplied them by 100, which we specified in Appendix B.2 as an implementation detail. \
> \
> Passing the gradients only through the most salient filters is another interesting way of constructing the input-space saliency, however, we specifically chose to use a similarity/distance metric to constrain the saliency of the rest of the filters to not change thus isolating the pixels that affect solely the chosen filters.

---

> ### Author Response · Authors · 2021-11-22
> **Part 3: Addressing the detailed comments on Experiments and General**
>
> **Experiments**
>
> * *“Section 3.1/Figure 3: If I understand it correctly these plots only contain data from initially misclassified samples. While the evaluation is done over all misclassifications, it is the sum of experiments on single images. I.e. when a filter is removed this is done only for a single example and evaluated on this example. In that sense it is similar to an adversarial perturbation in the middle of the network. The fact that this changes the confidence should not be surprising per se.”* \
> \
> As we indicate in the paper, the experiment simply serves as a sanity check which validates that the method does what one would expect and want and indeed finds the parameters most responsible for misclassification. We entirely agree with you that the result is unsurprising.
>
>
> * *“Section 3.2/Figure 4/5: The nearest neighbor approach is done by computing the cosine similarity of the saliency maps. Since this is dominated by the higher layers it should not be surprising that this results in conceptual similarity vs input similarity. This can also explain why the same classes are misclassified since higher level concepts are also higher in the layers. In Fig. 5 similarity of low vs higher level layers is shown. The texts states that this is done to show misbehavior, but it is not clear what is misbehaving in this plot.”*\
> \
> We agree that the nearest neighbor approach likely has a relationship to feature space similarity which would explain why it indicates conceptual similarity, although similarity in saliency space does not yield the same neighbors as similarity in feature space, as shown in our new experiments in section A.2 (Figure 11).\
> \
> Consider that an image from class A that is misclassified into class B because filter k failed to extract a discriminatory feature will often get matched, using our saliency profiles, with an image from class B that is misclassified into class A because the same filter k failed to extract discriminatory features.  Thus, because a small number of salient filters dominate the saliency profiles, these profiles match samples which a model misclassified as a result of the same network components, even if these samples are not in the same class (either in the sense of ground-truth or the model’s predictions). We concretely observe this phenomenon in Figure.4.  Our metric doesn’t only favor later layers in general but specifically favors parameters with large gradients and computes magnitudes rather than signed values. This intuition, in which samples are matched via common model component failures, underpins our statement regarding “misbehavior”.  We have updated our draft to disambiguate this statement, and we thank you for pointing out this confusion.
>
>
> * *“Section 3.3 Correcting mistakes by updating salient filters is evaluated on all images independently. How was batch norm used in this case for the update? Setting it to training mode could have weird effects. I am not convinced that it should be surprising that updating the ones with large gradients results in changes at the output.”* \
> \
> We set batch normalization to test mode when we update salient filters to avoid the problem you identify.  We have now updated our draft to clarify.\
> \
> Section 3.3 investigates how fine-tuning the most salient filters on a single image affects the nearest neighbors of that image in the saliency space. Panels (b) and (c) in Figure 6 suggest that the nearest neighbors found using our method are images that are wrong for similar enough reasons that they can be corrected by only updating the most salient filters on a single image. Also, our additional results in section A.3 suggest that this effect can generalize to other datasets.\
> \
> We would also like to mention that in Section B.1, we have included the details of updating the chosen filters, which is done by taking a step of fixed size regardless of their original gradient magnitude, using only the sign of the gradients. Therefore, salient filters are not updated by a larger amount than random or the least salient ones.
>
>
> * *“Section 3.4 This section is the most problematic to me since it lacks large scale evaluation and is mostly anecdotal.”*\
> \
> Please, see Part 1 of our response.
>
>
> **General**
> * *“Make the labels of the axes in plots larger. They are hard to read.”* \
> \
> Thank you, we made the axis labels bigger.

---

> > ### Author Response · Authors · 2021-11-30
> > **Thank you for your review**
> >
> > Dear Reviewer tkQp,
> >
> > Thank you again for your feedback and suggestions on improving our paper. We hope that our clarifications, additional experiments, and improved visualizations address your concerns. We would very much appreciate it if you considered increasing the score of our paper.
> >
> > Thank you!

---

> > > ### Comment · Reviewer_tkQp · 2021-11-30
> > > **Thanks for the clarifications**
> > >
> > > The clarifications helped to understand the paper better.
> > > I am happy to see that the authors found the remarks and the suggestion helpful to find ways to improve the paper.
> > >
> > > Unfortunately given that one of the key issues: section 3.4 being quite anecdotal cannot be resolved in time I would not update the score.

---

> > > > ### Author Response · Authors · 2021-12-02
> > > > **We Are Incorporating Your Feedback**
> > > >
> > > > We would like to emphasize that our work already contains large-scale evaluations of our parameter-saliency method in other sections, while Section 3.4 is intended to be a how-to guide for practitioners that read our paper and want to see an example of how to use our method. We believe that from this perspective, Section 3.4 is valuable.
> > > >
> > > > Nonetheless, we appreciate your feedback and agree that performing the case study on a large scale would be valuable.  We have started running experiments to address these suggestions, but since the emergency review was published halfway through the rebuttal process, we were extremely limited in time. The Causal ImageNet dataset with target object masks turned out to not be publicly available, and we are instead using the ImageNet bounding box annotations. Since the bounding boxes only exist for the ImageNet train set, we must split the training set and train models from scratch on the training split.  We will include the results of the experiments we are running now in the camera-ready version of our paper.  We want to point out that we ran numerous experiments and updated the draft accordingly prompted by the feedback from the non-emergency reviewers, and we are thoroughly incorporating your feedback as well.
> > > >
> > > > Thank you!

---

### Author Response · Authors · 2021-11-29
**Follow up and thank you**

Dear Reviewers,

Thank you for taking the time to review our work and for your thoughtful feedback. We appreciate all the positive comments and are encouraged that the reviewers found our perspective of identifying and analyzing parameters responsible for misclassification interesting (Reviewer 3o7Z and Reviewer qR4u) and novel (Reviewer qR4u), and the findings of our paper interesting (Reviewer 3o7Z) and inspiring (Reviewer qR4u). We also appreciate that the reviewers found our motivation straightforward and easy to understand (Reviewer sCeW), our analysis thorough and with practical potential (Reviewer 3o7Z) and that our contributions are verified by the proposed experiments (Reviewer qR4u).

Nevertheless, some concerns were raised during the rebuttal, which we address in our individual responses to the reviewers. We have also uploaded a revised version of our paper with the results of our new experiments prompted by the reviews.
In particular, we have added new experiments on:

* Comparison to adversarial attacks in parameter space (to address the feedback of Reviewer 3o7Z and Reviewer tkQp).
* Random perturbations as an alternative to reveal sensitivity of salient parameters (as requested by Reviewer 3o7Z)
* Extending our fine-tuning experiments to independent datasets (as requested by Reviewer sCeW)
* Applying our parameter saliency method to language models to showcase the utility of our method beyond image classification (as asked by Reviewer sCeW)

Additionally, we are working on scaling up our experiments from section 3.4 as requested by Reviewer tkQp in the emergency review and we will include the results in the camera-ready version of the paper.

Thank you!

---

### Decision · Program_Chairs · 2022-01-20

**Decision:**

Reject

**Comment:**

For this paper initially the reviews were 6,8,5,5. All the reviewers have provided constructive and substantial feedback. The authors have incorporated changes to address some of these comments and some of the comments could not be addressed. The main criticism of the reviewers have been that the Reviewer tkQp finds two clear limitations in the paper, Reviewer 3o7Z finds that the proposed idea is similar to the parameter-space adversarial attacks and Reviewer sCeW questions the generalisability of the method to other tasks. After the rebuttal the reviewers have reached the consensus that the paper may not be above the acceptance threshold (final scores: 6,6,5,5). Following the reviewers' recommendations, the meta reviewer recommends rejection.